# Endocannabinoid and Nitric Oxide-Dependent IGF-I-Mediated Synaptic Plasticity at Mice Barrel Cortex

**DOI:** 10.3390/cells11101641

**Published:** 2022-05-14

**Authors:** José Antonio Noriega-Prieto, Laura Eva Maglio, Sara Ibáñez-Santana, David Fernández de Sevilla

**Affiliations:** 1Departamento de Anatomía, Histología y Neurociencia, Facultad de Medicina, Universidad Autónoma de Madrid, 28029 Madrid, Spain; norie015@umn.edu (J.A.N.-P.); lauemaglio@gmail.com (L.E.M.); sara.ibannezs@estudiante.uam.es (S.I.-S.); 2Department of Neuroscience, University of Minnesota, Minneapolis, MN 55455, USA

**Keywords:** IGF-I, endocannabinoids, nitric oxide, spike timing-dependent plasticity

## Abstract

Insulin-like growth factor-I (IGF-I) signaling plays a key role in learning and memory. IGF-I increases the spiking and induces synaptic plasticity in the mice barrel cortex (Noriega-Prieto et al., 2021), favoring the induction of the long-term potentiation (LTP) by Spike Timing-Dependent Protocols (STDP) (Noriega-Prieto et al., 2021). Here, we studied whether these IGF-I effects depend on endocannabinoids (eCBs) and nitric oxide (NO). We recorded both excitatory postsynaptic currents (EPSCs) and inhibitory postsynaptic currents (IPSCs) evoked by stimulation of the basal dendrites of layer II/III pyramidal neurons of the Barrel Cortex and analyzed the effect of IGF-I in the presence of a CB_1_R antagonist, AM251, and inhibitor of the NO synthesis, L-NAME, to prevent the eCBs and the NO-mediated signaling. Interestingly, L-NAME abolished any modulatory effect of the IGF-I-induced excitatory and inhibitory transmission changes, suggesting the essential role of NO. Surprisingly, the inhibition of CB1Rs did not only block the potentiation of EPSCs but reversed to a depression, highlighting the remarkable functions of the eCB system. In conclusion, eCBs and NO play a vital role in deciding the sign of the effects induced by IGF-I in the neocortex, suggesting a neuromodulatory interplay among IGF-I, NO, and eCBs.

## 1. Introduction

Insulin-like growth factor I (IGF-I) is a peptide involved in the neuroplasticity of the Central Nervous System (CNS), being essential in learning and memory processes [1,2,3,4,5]. Indeed, IGF-I increases neuronal firing [6,7,8] and modulates synaptic transmission in many brain areas [9,10,11,12]. In the barrel cortex, an area of the somatosensory cortex that controls tactile information processing coming from the vibrissae [13], IGF-I levels modulate synaptic plasticity in rodents [14]. Somatosensory cortex astrocytes respond to sensory stimuli and regulate the sensory-evoked neuronal network [15]. Moreover, astrocytes are required for the IGF-I-mediated synaptic plasticity at both excitatory and inhibitory synapses [16]. We have demonstrated that the activation of IGF-I receptors (IGF-IRs) in astrocytes triggers the calcium-dependent release of ATP/Ado and the activation of A2A adenosine receptors that results in the long-term depression (LTD) of the inhibitory synapses. In addition, the ATP/Ado also induced a short-term increase in the efficacy of the excitatory synaptic transmission [16]. This IGF-I-mediated plasticity controls the threshold of the Hebbian synaptic plasticity induced by spike timing-dependent plasticity (STDP) in the mice barrel cortex [16].

The endocannabinoid system (eCBs) is a group of fatty acids that encompasses anandamide (AEA) and 2-arachidonoylglycerol (2-AG), the best characterized eCBs. These are synthesized in the postsynaptic neurons in a calcium-dependent manner regulated by neuronal activity and released, acting, mainly, in a retrograde manner. They carry out a plethora of physiological functions, such as synaptic plasticity, behavior, and emotional memories, by two G-protein-coupled receptors (GPCR) and cannabinoid receptor type-1 and cannabinoid receptor type-2 (CB1R and CB2R) expressed in neurons and glial cells. These receptors are distributed along the central and peripheral nervous system highlighting the key role of the endocannabinoid system in the neuronal function and information processing [17,18,19,20,21,22,23,24].

As we mentioned above, it is well known that postsynaptic activity and cytosolic Ca^2+^ increases in the pyramidal neurons (PNs) are of key importance in the synthesis and release of nitric oxide (NO) and endocannabinoids (eCBs), retrograde messengers that are implicated in the induction of short- and long-term plasticity at both excitatory and inhibitory synapses [25,26]. Although IGF-I regulates both excitatory and inhibitory synaptic transmission at the barrel cortex [16], the involvement of endocannabinoids and NO in this IGF-I-mediated modulation has not been studied.

Here, we analyzed the effect of NO and eCBs in the synaptic plasticity induced by IGF-I and in the facilitation of the long-term potentiation (LTP) induced by STDP (t-LTP). We have studied the effect of IGF-I on both EPSCs and IPSCs in the presence of a cannabinoid type 1 receptor (CB1R) antagonist, AM251, and of an inhibitor of the NO synthesis, L-NAME, to prevent the eCB- and NO-mediated signaling. Interestingly, the inhibition of the NO synthase prevented both the modulation of EPSCs and IPSCs by IGF-I, suggesting that the release of NO is required for both forms of synaptic plasticity. Moreover, although the inhibition of CB1Rs did not alter the modulation of IPSCs by IGF-I, it reversed the modulation of the EPSCs from a potentiation to a depression and prevented the IGF-I-mediated facilitation of the t-LTP. Overall, our findings provide evidence that both eCBs and NO determine the sign of the synaptic plasticity induced by IGF-I in the barrel cortex.

## 2. Materials and Methods

### 2.1. Ethics Statement and Animals

All animal procedures were approved by the Ethical Committee of the Universidad Autónoma of Madrid with Spanish (R.D. 1201/2005) and European Community Directives (86/609/EEC and 2003/65/EC), which promote the animal welfare. Male mice C57BL/6J (15–21 days old) were housed under a 12 h/12 h light/dark cycle with up to five animals per cage and were used for slice electrophysiology.

### 2.2. Slice Preparation

Briefly, C57BL6/J male mice were decapitated, and brains were removed and submerged in cold (4 °C) artificial cerebrospinal fluid (ACSF) containing (in mM): 124.00 NaCl, 2.69 KCl, 1.25 KH_2_PO_4_, 2.00 Mg_2_SO_4_, 26.00 NaHCO_3_, 2.00 CaCl_2_, and 10.00 glucose. The pH was 7.3–7.4. Coronal slices (350 μm thick) were cut with a Vibratome (Leica VT 1200S) and incubated in the ACSF (>1 h, at room temperature, 25–27 °C). Slices were transferred to a chamber (2 mL/min) superfused at room temperature with carbogen-bubbled ACSF (95% O_2_, 5% CO_2_) and fixed to an upright microscope stage (BX51WI; Olympus, Tokyo, Japan) equipped with infrared differential interference contrast video (DIC) microscopy and a 40× water-immersion objective. In some experiments, L-NAME (50 µM) or AM-251 (10 µM) was added to the ACSF to block nitric oxide synthetase (NOS) or CB1R, respectively. 

### 2.3. Electrophysiological Recordings

Patch-clamp recordings of layer II/III pyramidal neurons (PNs) of the barrel cortex were performed in the whole-cell voltage-clamp and current-clamp configurations with patch pipettes (4–8 MΩ) filled with an internal solution that contained (in mM): 112.5 Cs-Gluconate, 20 HEPES-K, 0.2 EGTA, 4 Na_2_-ATP, 0.3 Na_3_-GTP, 8 CsCl, pH 7.2–7.3 (~280 mOsm), pH 7.2–7.3. Recordings were performed using a Cornerstone PC-ONE amplifier (DAGAN, Minneapolis, MN, USA). Pipettes were placed with a mechanical micromanipulator (Narishige, Tokyo, Japan). The holding potential was adjusted to −60 mV, and the series resistance was compensated to ~80%. Layer II/III PNs were accepted only when the seal resistance was >1 GΩ, and the series resistance (10–20 MΩ) did not change (>20%) during the experiment. Data were low-pass-filtered at 3.0 kHz and sampled at 10.0 kHz, through a Digidata 1500A (Molecular Devices, Sunnyvale, CA, USA). Synaptic responses were evoked with Pt/Ir concentric bipolar (OP 200 μm, IP 50 μm, FHC) stimulating electrodes placed at layer IV and connected to a Grass S88 stimulator and stimulus isolation unit (Quincy, MA, UDA). Single pulses (100 μs duration and 20–100 µA) were continuously delivered at 0.33 Hz. IGF-I-mediated modulation of the EPSCs and IPSCs was analyzed in the same cell by recording the EPSCs at −60 mV for 3 min followed by the IPSCs at 0 mV for 30 s. After 14 min of stable currents, the recording was switched to the current clamp mode to monitor the postsynaptic potentials (PSPs) at 0.2 Hz. To test whether IGF-I modifies the number of synaptic stimuli that evoked APs in the recorded PN, the stimulation intensity was increased to reach ≈10% of the suprathreshold responses for 5 min. These suprathreshold responses did not change either the synaptic currents or the number of synaptic stimuli that evoked Aps in the absence of IGF-I (Appendix A). Then, IGF-I was added to the bath. Finally, after 15 min of IGF-I, the recording was switched back to the voltage-clamp, the stimulation intensity was readjusted to control values, and EPSCs and IPSCs were continuously monitored after IGF-I was washed out. Although it is known that cesium blocks chloride ions extrusion via the KCC2 cotransporter [27,28], thus leading to I–V plots in which the reversal potential of the GABAergic currents is equal to the holding potential, this effect has no significance on the main conclusions of the present study. Indeed, we have previously published a similar modulation of the GABAergic inhibition by IGF-I using a potassium methyl sulfate-based intracellular solution [16]. In some cases, to monitor the role of endocannabinoids in the t-LTP facilitated by IGF-I [16], the internal solution contained (in mM): 135 KMeSO_4_, 10 KCl, 10 HEPES-K, 5 NaCl, 2.5 ATP-Mg^+2^, and 0.3 GTP-Na^+^, buffered to pH 7.3.

### 2.4. Data Analysis

The pre- or post-synaptic origin of the synaptic plasticity induced by IGF-I observed was tested by estimating changes in the paired-pulse ratio (PPR) of the EPSCs and IPSCs [29,30,31]. To estimate the modification of the synaptic current variance induced by IGF-I, the noise-free coefficient of variation in the synaptic responses before and during IGF-I was calculated [31]. Data analysis was performed in Clampfit 10 (Axon Instrument) and graphs were drawn in SigmaPlot 11. Data normality was tested using a Shapiro–Wilk test. Statistical estimates were made with Student’s two-tailed *t*-tests for unpaired or paired data as required and one-way ANOVA with post hoc Holm–Sidak or Kruskal–Wallis tests. Data are presented as means ± SE. The threshold for statistical significance was *p* < 0.05 (*), *p* < 0.01 (**), and *p* < 0.001 (***) for Student’s *t*-test; and *p* < 0.05 (#), *p* < 0.01 (##), and *p* < 0.001 (###) for one-way ANOVA.

## 3. Results

### 3.1. IGF-I Modulates Synaptic Transmission

We studied the effect of IGF-I on the excitatory and inhibitory synaptic transmission recorded simultaneously in the same layer II/III PNs. For this purpose, we recorded the excitatory and inhibitory postsynaptic currents (EPSCs and IPSCs, respectively) at their reversion potentials (IPSCs were recorded at 0 mV and EPSCs were recorded at −60 mV Figure 1 and Appendix A) using a cesium-based internal solution (see methods). Under these conditions, IGF-I induced LTD of the IPSCs (from 100.60 ± 2.07 to 55.50 ± 5.31% of peak amplitude in control and during IGF-I, recorded at 0 mV; Figure 1A) and a short-term potentiation (STP) of the EPSCs (from 99.55 ± 1.90 to 141.78 ± 6.87% of peak amplitude in control and during IGF-I, recorded at −60 mV; Figure 1A). The IGF-I-mediated modulation of the EPSCs and IPSCs was prevented by NVP-AEW541 (400 nM; Figure 1A). Moreover, IGF-I increased the number of synaptic stimuli that evoked APs (from 8 ± 2.57 to 30.5 ± 8.43 number of APs; Figure 1B) and this effect was also prevented by NVP-AEW 541 (Figure 1B, bottom). In some experiments, IGF-I was not added, and the synaptic transmission and number of synaptic stimuli that evoked APs were not altered (Appendix A).

To analyze the locus of expression of both IGF-I-mediated synaptic plasticity, the pre- or postsynaptic origin of the effect of IGF-I was studied by constructing 1/CVr2 plots (see experimental procedures). We observed that the increase in the mean EPSC peak amplitude was not paralleled by an increase in the 1/CVr2 parameter (Figure 1C), which discards a pre-synaptic origin in the EPSC potentiation. Indeed, experimental 1/CVr2 values follow the predicted relationship for a postsynaptic site of action (horizontal line), which suggests that the potentiation of the EPSCs induced by IGF-I was mediated by a postsynaptic mechanism. Moreover, the experimental 1/CVr2 values for the IGF-I-mediated modulation of the IPSCs follow the diagonal line, which suggests a pre-synaptic locus of expression (Figure 1C). In addition, we recorded the EPSCs and IPSCs evoked by paired-pulse stimulation (50 ms delay) and analyzed the changes in the paired-pulse responses. The paired-pulse ratio (PPR) of the response was the quotient of the peak amplitude of the second over the first response. The PPR did not change during the IGF-I-mediated STP of the EPSCs (from 1.35 ± 0.10 to 1.31 ± 0.09 in control and during IGF-I; Figure 1D), indicating a postsynaptic mechanism. However, the PPR was increased during the IGF-I-mediated LTD of the IPSCs (from 1.16 ± 0.05 to 1.47 ± 0.08 in control and during IGF-I; Figure 1D), indicating a presynaptic mechanism. Taken together, these results indicate that IGF-I induces an increase in the spiking activity of layer II/III PNs and the simultaneous modulation of the excitatory and inhibitory contacts at the same neurons in this layer. These synaptic modulations are caused by the increase in the postsynaptic response on the excitatory spines to the released glutamate and by the reduction in the release of GABA from the inhibitory terminals.

### 3.2. The Induction of IGF-I-Mediated Synaptic Plasticity at Both Excitatory and Inhibitory Synapses Requires the Nitric Oxide Synthesis

It is well known that postsynaptic spiking activity and cytosolic Ca^2+^ levels are of key importance in the synthesis and release of NO and eCBs, retrograde messengers implicated in the induction and modulation of synaptic transmission and plasticity at both excitatory and inhibitory synapses [25,26,32]. Therefore, we perform experiments in the presence of L-NAME (50 µM), a NO synthetase inhibitor to determine whether NO was involved in the IGF-I-mediated modulation of both EPSCs and IPSCs. Both the IGF-I-mediated LTD of the IPSC and STP of the EPSC were prevented (Figure 2). However, the increase in the spiking activity of the PNs mediated by IGF-I in L-NAME (from 5.2 ± 0.86 to 28.4 ± 3.20 number of APs in control and after 15 min with IGF-I, Figure 2B) was similar to the recorded in ACSF (Figure 1B), indicating that this increase was not dependent on the modulation of synaptic transmission and pointed to a direct effect of IGF-I on the intrinsic properties of the PNs. These results indicate that NO is required for both the IGF-I-mediated IPSC and EPSC modulations, but not for the increase in the spiking activity of the PNs. 

### 3.3. Endocannabinoids Determine the Sign of the IGF-I-Mediated Synaptic Plasticity at Excitatory Synapses

As mentioned above, we also checked the role of eCBs on the synaptic plasticity mediated by IGF-I. For this purpose, we analyzed the effects of AM251 (10 µM), a CB1R antagonist, in the plasticity of both the EPSC and IPSC induced by IGF-I. Although the IGF-I-mediated LTD of the IPSCs was still observed in the presence of AM251 (from 99.05 ± 1.20 to 50.63 ± 4.40% of peak amplitude; Figure 3, the potentiation of the EPSCs was not only prevented but was reversed to an LTD (from 99.18 ± 1.20 to 50.85 ± 4.64% of peak amplitude; Figure 3B). This IGF-I-mediated depression of the EPSCs recorded in the presence of AM251 had a presynaptic locus of expression as there was a positive correlation between the 1/CVr^2^ values and the EPSC depression (Figure 3C). Interestingly, IGF-I did not increase the spiking activity of the recorded PN under AM251 (Figure 3B), suggesting that the modulation of the intrinsic properties of the PN depends on the activation of CB_1_Rs. Taken together, these results demonstrate that the eCBs are required for both the IGF-I-mediated increase in the spiking activity of the recorded PN and the EPSC potentiation, being unnecessary for the IPSC depression. Indeed, the sign of the IGF-I-mediated EPSC plasticity, potentiation or depression, depends on whether CB_1_Rs are activated or not, adding degrees of freedom to the modulation of synaptic transmission by IGF-I.

### 3.4. IGF-I-Mediated Facilitation of the t-LTP Requires CB_1_R Activation 

In order to check the functional impact of this EPSC dual modulation by IGF-I depending on the activation of CB_1_Rs, we analyzed the effect of AM251 in the IGF-I-mediated facilitation of the t-LTP that we have previously described [16]. We used STDP protocols consisting of a subthreshold PSP followed by a back-propagating action potential (BAP) at delays of 10 ms repeated 10, 20, and 50 times at 0.2 Hz (Figure 4). Under the control condition, 10 and 20 pairings were unable to induce the t-LTP of the PSPs. However, 50 pairings were sufficient to generate t-LTP of the PSPs (Figure 4B control). Similarly, we performed the experiments in slices where IGF-I was previously bath-applied and washed out (Figure 4A bottom). In this case, although 10 pairings were still unable to induce any modulation of the PSPs, 20 and 50 generated t-LTP (20 and 50 pairings; Figure 4B, IGF-I). These results suggest that the threshold to induce t-LTP had been facilitated by IGF-I. However, this facilitation was absent in similar experiments in which IGF-I was applied in the presence of AM-251 (Figure 4C). Indeed, under these conditions, neither 10 nor 20 pairings were sufficient to induce the t-LTP (10 pairings: from 99.49 ± 1.19 to 99.81 ± 3.16% of peak amplitude, 20 pairings: from 98.93 ± 0.84 to 97.15 ± 2.51% of peak amplitude, Figure 4D). Unexpectedly, 50 pairings were not only unable to produce t-LTP but induced an t-LTD of the PSPs (from 99.31 ± 0.86 to 78.06 ± 5.58% of peak amplitude; Figure 4E), which was absent when the PNs were not previously exposed to IGF-I (from 100.08 ± 0.93 to 149.85 ± 17.4% of peak amplitude; Figure 4E). These results suggest that, under AM-251 conditions, IGF-I can change the temporal coincidence rules for the t-LTP. Therefore, the dual modulation of the EPSCs by IGF-I, depending on the activation of CB_1_Rs, may serve as a switch in the temporal coincidence rules of the Hebbian synaptic plasticity, thus having a great functional impact in the information processing at the barrel cortex. 

## 4. Discussion

In this article, we present new data clarifying the mechanisms implicated in the modulation of synaptic transmission by IGF-I at the barrel cortex. Here, we show that IGF-I mediated an increase in PN spiking activity and both the modulation of the glutamatergic and GABAergic synaptic transmission depending on the release of NO and eCBs. In this scenario, the increase in PN spiking activity would lead to an increase in cytosolic Ca^2+^ levels and therefore the release of NO and eCBs. NO is a well-known modulator of neuronal function that is able to enhance excitability by reducing the GABA-mediated Cl- influx [33] and by increasing the probability of glutamate release [34,35,36]. Moreover, other studies have shown that NO mediates an NMDA receptor-independent LTP [37]. As we have demonstrated here, the release of NO is mandatory to induce the IGF-I-mediated EPSCs potentiation at layer II/III PNs. 

Likewise, it has been widely established the modulation of synaptic transmission by eCBs [22,38]. In fact, the inhibition of GABAergic transmission through the activation of CB_1_Rs in GABAergic terminals is a well-known mechanism that leads to the enhancement of glutamatergic synaptic transmission [38]. In layer II/III, CB_1_Rs are primarily located at GABAergic interneurons [39], while it has been suggested that they are absent in glutamatergic presynaptic terminals at this layer. Therefore, eCBs trigger an increase in excitatory transmission by inhibiting the GABA release from interneurons [40]. Our results demonstrate that the release of eCBs is needed for the EPSCs potentiation, probably by decreasing the GABAergic inhibitory transmission. 

Another possible scenario involving both NO and eCBs is deduced from recent evidence on the contribution of astrocytes as key modulators in synaptic transmission and plasticity [41]. In fact, it has been demonstrated that eCBs potentiate synaptic transmission through the activation of astrocytic CB_1_Rs [42,43] and that the coincidence of eCBs signaling and postsynaptic activity leads to an eCB-induced long-term potentiation (eLTP) [43] in hippocampal glutamatergic synapses. In this later form of LTP, activity-dependent depolarization of the postsynaptic neuron produces the release of NO and eCBs. While eCBs activate astrocytic CB_1_R, inducing the release of glutamate from the astrocyte and activation of presynaptic group I metabotropic glutamate receptors (mGluRs), NO activates protein kinase C (PKC) at the presynaptic terminal. Both the activation of presynaptic mGluR type I receptors and PKC produced the increase in the release of glutamate from the presynaptic neuron, leading to the potentiation of excitatory transmission [43]. Therefore, we cannot rule out a possible contribution of astrocytes to the modulation induced by IGF-I described here. 

The STP of the EPSC is postsynaptically mediated and dependent on the release of both eCBs and NO, whereas the LTD of the EPSC is presynaptically mediated and dependent only on the release of NO. A recent study suggests that presynaptic IGF-IRs are basally active, thus regulating glutamatergic synaptic transmission by modulating the glutamate release probability [7]. In fact, it has been suggested that the tonic release of IGF-I and subsequent activation of IGF-IRs modulate the synaptic vesicle release, producing a short-term depression in excitatory hippocampal neurons [7]. Considering that IGF-I-induced STD of the ESPCs is presynaptically mediated and its induction does not require postsynaptic activity or cytosolic calcium increases, our results strongly suggest that the tonic modulation of the synaptic transmission induced by IGF-I described at the hippocampus may be still present at glutamatergic synapses of the barrel cortex. 

### Functional Role of the Modulation of Synaptic Transmission by IGF-I

We show that the IGF-I may increase or decrease the threshold of the t-LTP of EPSPs depending on the activity of CB_1_Rs. Interestingly, several processes triggered by IGF-I converge to cooperate in this action. First, the EPSC potentiation boosted by the IPSC depression induced by IGF-I in the absence of AM251 would enhance spine depolarization that occurred during the AMPAR-mediated transmission, as well as the backpropagation of the APs (BAPs). This temporal coincidence would be crucial to allow the necessary Ca^2+^ influx to induce t-LTP by increasing the relief of the voltage-dependent Mg^+2^ blockade of NMDARs. Therefore, the NMDAR component would amplify the depolarization and duration of the EPSP, and thus contribute to the intracellular Ca^2+^ increase by further expanding Ca^2+^ influx. This higher level of intracellular Ca^2+^ concentration would explain the requirement of fewer pairings between the EPSP and the BAPs to induce a similar t-LTP. However, in the presence of AM251, IGF-I would reduce the excitatory synaptic transmission to a level in which the induction of t-LTP is impaired even in a condition of a decreased inhibitory synaptic transmission. 

In summary, our data show that IGF-I regulates the intrinsic properties of layer II/III PNs and local excitatory-inhibitory circuits, which could explain the IGF-I effects on cortical activity. These novel results suggest that IGF-I plays an essential function in the induction of synaptic plasticity at the barrel cortex. Our findings support a possible major neuromodulatory action of IGF-I with a high physiological relevance of these phenomena on synaptic activity, which could have relevant implications for predicting how neuronal behavior might organize sensory processing in the barrel cortex. 

## Figures and Tables

**Figure 1 cells-11-01641-f001:**
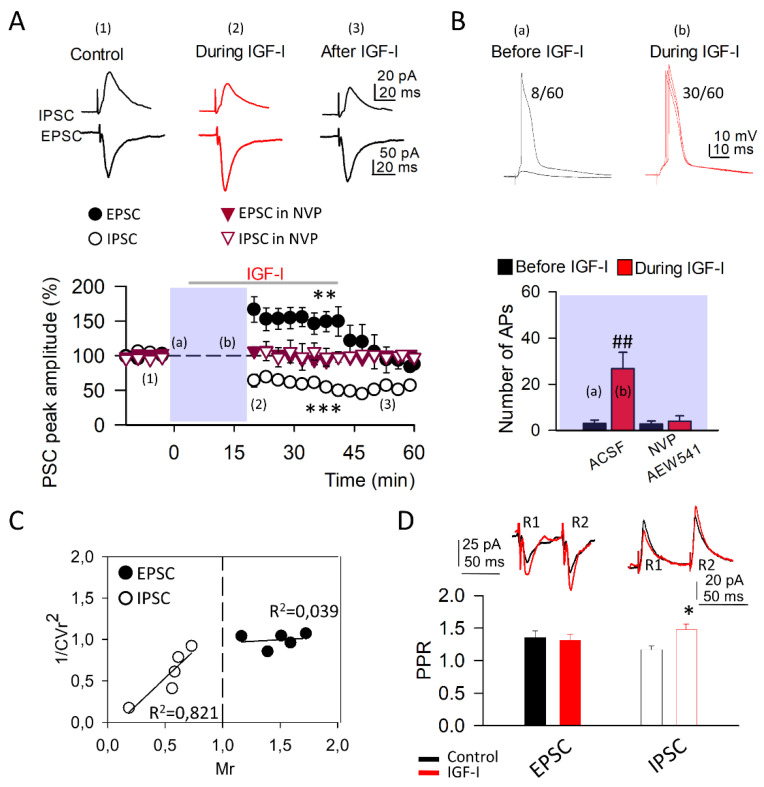
IGF-I modulates both excitatory and inhibitory synaptic transmission. (**A**) *Top*, Representative inhibitory and excitatory postsynaptic current traces (IPSC at 0 mV and EPSC at −60 mV, respectively) recorded from layer II/III pyramidal neurons in control conditions (left, black, (1)), during 10 nM IGF-I application (middle, red, (2)), and after IGF-I application (right, black (3)). *Bottom*, Time course of IGF-I effects on IPSCs (empty symbols; n = 8 ACSF vs. IGF-I *** *p* < 0.001) and EPSCs (full symbols; n = 8 ACSF vs. IGF-I *** *p* < 0.001) in the absence (black circles) and presence (red triangles of the IGF-I receptor antagonist NVP (400 nM; NVP vs. NVP + IGF-I, n = 5, *p* = 0.234, (IPSC) and NVP vs. NVP + IGF-I, n = 5, *p* = 0.658 (EPSC)). IGF-I was applied for 35 min, after the recording of a 5-min stable baseline. (**B**) *Top*, Representative superimposed current-clamp traces recorded before (black, (a)) and during IGF-I application (red, (b)). Numbers indicate the number of evoked action potentials (8 vs. 30) out of the total number of recordings (60) from this representative experiment. *Bottom*, Number of action potentials before (black) and during IGF-I application (red). ## *p* < 0.01. (**C**) Plots of variance (1/CVr^2^) as a function of normalized mean peak EPSC (full circles; n = 5 same as A) or ISPC (empty circles; n = 5 same as A) amplitude values (Mr) obtained 30 min after IGF-I application. (**D**) *Top*, Superimposed representative pair-pulse IPSC and IPSC recorded before (black trace) and during IGF-I (red trace). *Bottom*, EPSC (full bars; ACSF vs. IGF-I, n = 8, *p* = 0.43) and IPSC (empty bars; ACSF vs. IGF-I, n = 8, * *p* < 0.05 pair-pulse ratio (PPR) before (black) and after IGF-I (red) application.

**Figure 2 cells-11-01641-f002:**
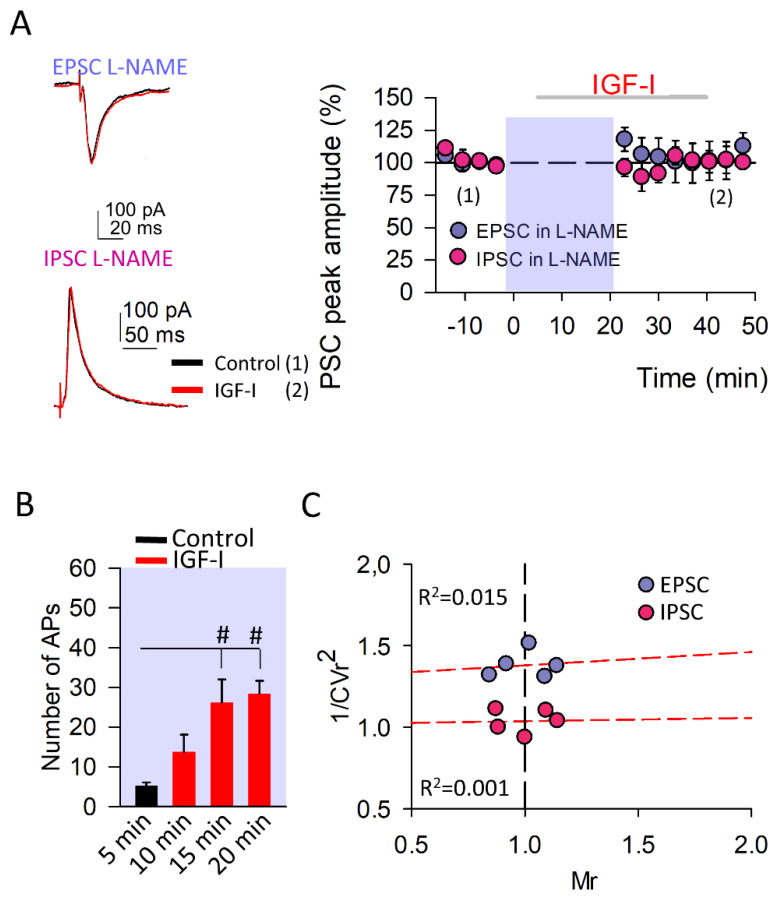
Synthesis of nitric oxide is required to induce synaptic plasticity at both excitatory and inhibitory synapses. (**A**) *Left*, Representative EPSC (at −60 mV) and IPSC (at 0 mV) recorded from layer II/III pyramidal neurons before (black traces (1)) and after IGF-I application (red traces (2)) in the presence of the nitric oxide synthase inhibitor L-NAME (50 µM). *Right*, Time course of IGF-I effects on EPSCs (purple circles; L-NAME vs. L-NAME + IGF-I, n = 5, *p* = 0.151) and IPSCs (pink circles; L-NAME vs. L-NAME + IGF-I, n = 5, *p* = 0.956) in the presence of L-NAME. IGF-I was applied for 35 min, after the recording of a 5-min stable baseline. L-NAME was applied through the recording pipette to inhibit postsynaptic nitric oxide synthases. (**B**) Number of action potentials during the baseline (control, black bar) and after 5, 10, and 15 min of starting IGF-I application (red bars; L-NAME vs. L-NAME + IGF-I, n = 5, # *p* < 0.05 for 15 and 20 min). (**C**) Plots of variance (1/CVr^2^) as a function of normalized mean peak EPSC (purple circles; n = 5 same A) or ISPC (pink circles; n = 5 same A) amplitude values (Mr) obtained 30 min after IGF-I application, when 50 µM L-NAME was applied through the recording pipette.

**Figure 3 cells-11-01641-f003:**
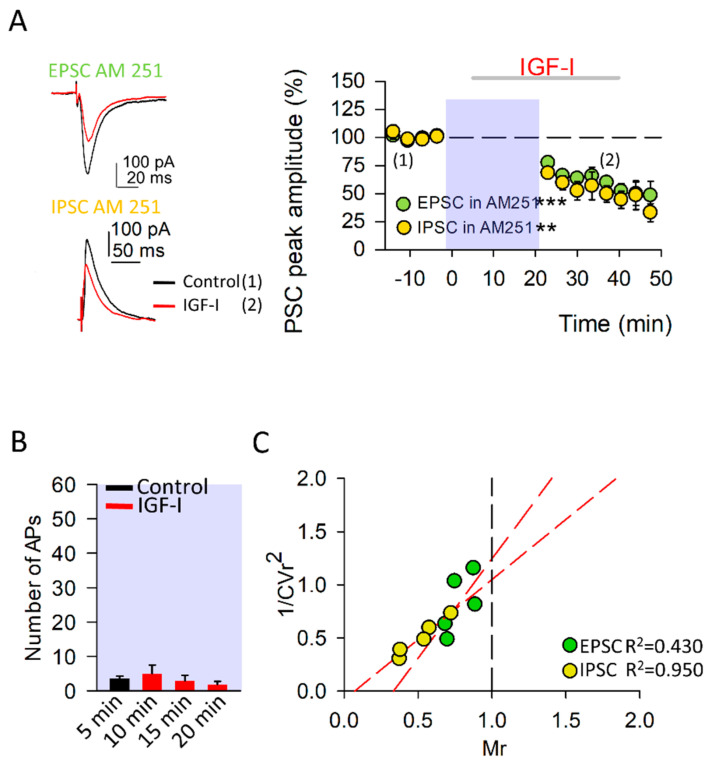
The endocannabinoids switch the IGF-I mediated synaptic plasticity at excitatory synapses. (**A**) *Left*, Representative EPSC (at −60 mV) and IPSC (at 0 mV) recorded from layer II/III pyramidal neurons before (black traces (1)) and after IGF-I application (red traces (2)) in the presence of the CB_1_R antagonist AM-251 (10 µM). *Right*, Time course of IGF-I effects on EPSCs (green circles; AM251 vs. AM251 + IGF-I, n = 5, *** *p* < 0.001) and IPSCs (yellow circles; AM251 vs. AM251 + IGF-I, n = 5, ** *p* < 0.01) in the presence of AM-251. IGF-I was applied for 35 min, after the recording of a 5-min stable baseline. (**B**) Number of action potentials during the baseline (control, black bar) and after 5, 10, and 15 minutes of starting IGF-I application (red bars; AM251 vs. AM251 + IGF-I, n = 5 *p* = 0.566). (**C**) Plots of variance (1/CVr^2^) as a function of normalized mean peak EPSC (green circles; n = 5 same A) or ISPC (yellow circles; n = 5 same A) amplitude values (Mr) obtained 30 min after IGF-I application in the presence of 10 µM AM-251.

**Figure 4 cells-11-01641-f004:**
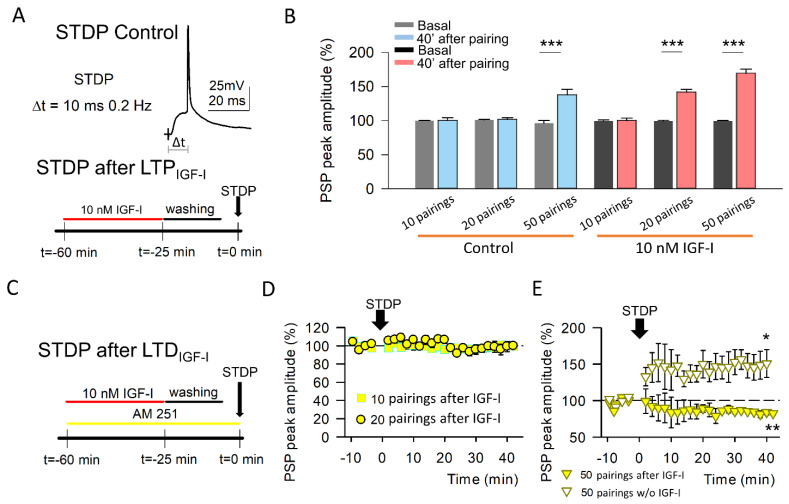
Activation of CB_1_R is required for IGF-I-mediated facilitation of the t-LTP. (**A**) *Top*, Representative traces obtained when applying the t-LTP induction protocol, where an evoked PSP is followed by an action potential (AP) with a 10 ms delay. *Bottom*, Scheme of the experimental approach used, where the t-LTP induction protocol was applied 60 min after the start of 10 nM IGF-I application (35 min of IGF-I application followed by 20 min of washout and a baseline recording for 5 min). (**B**) Normalized mean peak PSP amplitudes before (basal) and after 40 min of the application of the t-LTP induction protocol for 10, 20, and 50 PSP-AP pairings when the layer II/III pyramidal neurons were no exposed (control; 10 pairing, before vs after, n = 5 *p* > 0.05; 20 pairing, before vs after, n = 6 *p* > 0.05; 50 pairing before vs. after, n = 7 *** *p* < 0.001) or exposed to 10 nM IGF-I for 35 min before the t-LTP induction (10 pairing before vs after n = 5, *p* > 0.05; 20 pairing before vs after n = 6 *** *p* < 0.001; 50 pairing, before vs after n = 6, *** *p* < 0.001 (**C**) Scheme of the experimental approach used, where the t-LTP induction protocol was applied 60 min after the start of 10 nM IGF-I application (35 min of IGF-I application followed by 20 min of washout and a baseline recording for 5 min) in the continuous presence of 10 µM AM-251. (**D**) Time course of IGF-I pre-exposure effects on PSPs in the presence of AM-251 for 10 (squares; before vs. after n = 6, *p* = 0.912) and 20 (circles; before vs. after, n = 6 *p* = 0.532) PSP-AP pairings. (**E**) Time course of PSPs for 50 PSP-AP pairings when the layer II/III pyramidal neurons were no exposed (white triangles; before vs. after, n = 5 * *p* < 0.05) or exposed to 10 nM IGF-I (yellow triangles; before vs after, n = 5 * *p* < 0.05) for 35 min before the t-LTP induction in the continuous presence of 10 µM AM-251.

## Data Availability

Data are accessible upon personal request.

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
