# Peer review of "Endocannabinoid and Nitric Oxide-Dependent IGF-I-Mediated Synaptic Plasticity at Mice Barrel Cortex"

_cells, 2022, doi:10.3390/cells11101641_

Round 1

Reviewer 1 Report

In this work, Noriega-Prieto et al found a synaptic mechanism involved in plastic events mediated by the insulin-like growth factor-I (IGF-I). Overall, the manuscript is well written, and the results clearly explained.

The authors show that both endocannabinoid and nitric oxide signaling is recruited by IGF-I activity to elicit synaptic plasticity in the barrel cortex. These interesting results could open new venues of research aim to explore the same synaptic mechanisms in other brain regions (e.g., hippocampus) in vitro and in vivo behaving animals. Nevertheless, the data presented here is solid supporting the authors conclusion. Thus, I recommend this work to be published in Cells.

Author Response

We very much thank reviewer for the positive comments on the importance of our manuscript, and the constructive comments on the relevance of our results that could open new venues of research aim to explore the same synaptic mechanisms in other brain regions.

Reviewer 2 Report

Paper by Noriega-Prieto and coworkers investigated the role of IGF-I over excitatory and inhibitory innervation at the Barrel Cortex. Moreover authors explore the mutual interaction of IGF-I with eCBs and NO signaling. Authors find out an interesting effect of the IGF-1 in regulating  intrinsic excitability of Layer II-III neurons of Barrel cortex reporting also a clear dependence of IGF-1 effect by  the release of eCBs and NO.

Paper approach an interesting topic and data are well presented. However experimental approach rise some questions reducing the overall significance of the paper and that should be taken into account.

-  Authors investigated both excitatory and inhibitory input in the same cell by using a CS based internal solution. Although correct in principle, this approach require more details and a more accurate verification. Point of stimulation is the dame for both EPSCs and IPSCs. How authors can exclude a contamination of GABA and glutamatergic input during the recording.? Traces presented at 0mV are quite similar to an NMDA mediated current (slow rise and slow decay time). Did the authors tested this stimulation protocol in pharmacological isolation to avoid overlapping signaling?

Authors claim that the EPSCs and IPSCs have been recorded at their reversion potential (0mV and -40mV). This should be corroborated by references or by calculating the reversal potential of Cl- and Na2+for the internal solution used in the experiments

- Stimulation protocol to investigate the IPSCs and EPSCs (pg 3 line 96-98) is not clear. Is not clear the increase of stimulation intensity  for 5 minutes after the baseline. It look like a low-frequency stimulation protocol, but this should be better and clearly explained.

A CS-based solution block several ionic conductance largely involved in triggering and sustaining action potential. Thus CS-based solution is not the best choice to study the rate of evoked action potential. A K-based solution in current clamp configuration  should be preferentially used.

- Did the authors tested the effect of L-Name or AM251 per-sé without IGF-I incubation?

- PPR experiments should be report also the representative traces as all others experiments.

-Last lines of Abstract (18-22) are unclear and should be rephrased

Reviewer 3 Report

In this study authors evaluate the effect of IGF-I in synaptic plasticity and role of endocannabinoids and NO in this mechanism. The research is well studied and interesting. However, there are some concerns:

  1. Authors never talked about endocannabinoids in introduction. A brief introduction on that would be helpful for readers
  2. Endocannabinoids is complex system that works with the involvement of many components including, cannabinoid receptors (mainly CB1 and CB2), endogenous ligands that activate these receptors and the enzymes involved in their metabolism. It will be more valuable to look at the effects of CB1 agonist to see reverse effects authors shown in this study.
  3. I believe that rationale behind using CB1 antagonists and NO inhibitor is lacking. Does author tried to check the levels of CB1 receptors (whether it is activated) or level of key endocannabinoids and NO levels after IGF treatment in these animals?

Minor

  1. The abbreviation of EPSCs and IPSCs need to be mentioned in abstract.
  2. Lines 123-133 are very confusing. Sometimes, figures legends are mentioned in brackets or sometimes they are in main text. It needs to be more clearly explained.
  3. Number of n can be moved to figures legends or material and method instead of main text

Round 2

Reviewer 2 Report

Authors addressed my concerns. No further points required revision

Author Response

I would like to thank the reviewer.

Reviewer 3 Report

I appreciate the responses and efforts from the authors. The authors did a nice job in addressing my comments. I recommend accepting this manuscript.

Author Response

I would like to thank the reviewer.